

# 1 Assessing approaches to determine the effect of ocean

# 2 acidification on bacterial processes

**Tim J. Burrell**[1,2, †]**, Elizabeth W. Maas**[1,*]**, Paul Teesdale-Spittle**[2] **and Cliff S. Law**[1,3]
[1] {National Institute of Water and Atmospheric Research, Greta Point, Wellington, New
Zealand}
[2] {Victoria University of Wellington, School of Biological Sciences, Wellington, New
Zealand}
[†] {now at: C-MORE, University of Hawaii at Mānoa, Honolulu 96822, Hawaii}
[*] {now at: Ministry for Primary Industry, PO Box 12034, Ahuriri, Napier, New Zealand}
[3] {Department of Chemistry, University of Otago, Dunedin, New Zealand}
Correspondence to: T. Burrell (timbo.burrell@gmail.com)

## 13 Abstract

Bacterial extracellular enzymes play a significant role in the degradation of labile organic
matter and nutrient availability in the open ocean. Although bacterial production and
extracellular enzymes may be affected by ocean acidification, few studies to date have
considered the methodology used to measure enzyme activity and bacterial processes. This
study investigated the potential artefacts in determining the response of bacterial extracellular
glucosidase and aminopeptidase to ocean acidification, and the relative effects of three different
acidification techniques. Tests confirmed that the fluorescence of the artificial fluorophores
was affected by pH, and that addition of MCA fluorescent substrate alters seawater pH. In
experiments testing different acidification methods, bubbling with $CO_2$ gas mixtures resulted
in higher $\beta$-glucosidase activity relative to acidification by their introduction via gas-permeable
silicon tubing, or by acid addition (HCl). In addition, bacterial numbers were 15–40 % higher
with bubbling relative to seawater acidified with gas-permeable silicon tubing and HCl.
Bubbling may lead to overestimation of carbohydrate degradation and bacterial abundance, and
consequently incorrect interpretation of the impacts of ocean acidification on organic matter
cycling.



## 1 Introduction

Proteins and carbohydrates constitute two of the most common labile organic substrates in the ocean (Benner, 2002; Benner et al., 1992; McCarthy et al., 1996), both of which are essential for cellular growth and repair (Azam et al., 1983; Simon and Azam, 1989). Two groups of extracellular enzymes commonly studied for their role in protein and carbohydrate degradation are aminopeptidases and glucosidases, respectively. Enzyme activity is sensitive to different environmental factors, and consequently degradation of proteins and carbohydrates will vary accordingly. Most enzymes are pH sensitive and have different pH optima (Tipton and Dixon, 1979, Piontek et al., 2013), and consequently a change in ocean pH may result in a decline or increase in activity of extracellular enzymes as these are directly exposed to the external seawater pH (Orsi and Tipton, 1979; Tipton and Dixon, 1979). Atmospheric $CO_2$ has increased by 40 % since the 18$^{th}$ century (IGBP-IOC-SCOR, 2013; IPCC, 2013), which is of concern as $CO_2$ freely exchanges with the ocean and directly alters ocean carbonate chemistry and pH. As a result ocean pH has declined from 8.2 to 8.1, with a continued decline to 7.8 predicted by the year 2100. This decline in ocean pH and the associated change in carbonate chemistry, referred to as ocean acidification (OA), will significantly impact metabolic reactions and influence carbon cycling in the ocean (Endo et al., 2013; Engel et al., 2014; Piontek et al., 2010; Riebesell et al., 2007). For this reason, researchers have investigated the sensitivity of a wide range of biotic and abiotic factors to future changes in ocean pH and the carbonate system.

Bacterial extracellular enzyme activity has been investigated in OA studies (reviewed in Cunha et al., 2010) due to the important role they play in the degradation of labile high molecular weight organic matter (Azam and Ammerman, 1984; Azam and Cho, 1987; Law, 1980; Münster, 1991) and the vertical flux of carbon to the deep ocean (Piontek et al., 2010; Riebesell and Tortell, 2011; Segschneider and Bendtsen, 2013). Current research suggests that bacterial extracellular enzyme activities may increase under future OA conditions (Grossart et al., 2006; Maas et al., 2013; Piontek et al., 2010, 2013; Yague and Estevez, 1988). This may result from the direct effect of pH on the ionisation state of the enzyme's component amino acids (Dixon, 1953), or from indirect influences on longer timescales (Boominadhan et al., 2009). The latter may be arise in response to changes in the concentration and composition of high molecular weight organic substrate due to the effect of pH on phytoplankton and bacterioplankton community composition (Endo et al., 2013; Engel et al., 2008; Riebesell, 2004; Witt et al.,



2011), bacterial secondary production and cell numbers (Endres et al., 2014; Maas et al., 2013),
and phytoplankton-derived organic exudation (Engel, 2002; Engel et al., 2014).
Bacterial extracellular enzyme activity is regularly determined using artificial fluorogenic
substrates. These substrates consist of a fluorescent moiety covalently linked to one or more
natural monomer molecules (Arnosti, 2011; Kim and Hoppe, 1984). The molecule is non-
fluorescent until it is hydrolysed by an extracellular enzyme, which triggers a fluorescent
response, allowing it to be detected and quantified (Hoppe, 1993). The sensitivity of the
analytical method to pH has been assessed in terrestrial soils (Malcolm, 1983; Niemi and
Vepsäläinen, 2005), however limited information is available on how these components
respond to a reduction in seawater pH (Piontek et al., 2013). If pH does have a significant effect
on the individual assay components, and this is not corrected, then calculated enzyme kinetics
will under or overestimate the true activity rates.
Several methods are commonly used to artificially adjust seawater pH (Cornwall and Hurd,
2015; reviewed in Riebesell et al., 2010). The simplest acidification method involves the
addition of a strong acid (typically HCl). The acid decreases the sample pH through the
formation of hydronium ions and modifies total alkalinity (TA), but does not alter dissolved
inorganic carbon (DIC) in a closed system (Emerson and Hedges, 2008); consequently
although it is relatively simple to adjust pH using acid, the balance of carbonate species does
not reflect the changes that will occur in response to increased $CO_2$ uptake unless corrected for
by the addition of a base (Iglesias-Rodriguez et al., 2008; Riebesell et al., 2010). Another
method for acidifying seawater is the use of $CO_2$-Air gas mixtures, which alter the seawater
carbonate species in ratios predicted to occur from the uptake of atmospheric $CO_2$ under future
scenarios (Gattuso and Lavigne, 2009; Riebesell et al., 2010; Rost et al., 2008; Schulz et al.,
2009). Schulz et al. (2009) suggest that microbial organisms are likely to respond to changes
in carbonate species (e.g. $CO_2$, $HCO_3^-$ or $CO_3^{2-}$), rather than changes in overall DIC or TA. A
review by Hurd et al. (2009) concluded that differences in carbonate chemistry arising from
the use of different acidification methodologies can influence phytoplankton photosynthesis
and growth rates, as well as particulate organic carbon production per cell, and so it is important
to ensure changes in all carbonate system species reflect that projected from an increase in $CO_2$
(Cornwall and Hurd, 2015).



In addition to the method of acidification, the mode of application also needs to be considered.
A commonly used method of introducing $CO_2$-Air gas mixtures into seawater is by bubbling.
This method is simple to implement and maintain for extended periods, however, the physical
disturbance associated with bubbling $CO_2$ gas may influence coagulation of organic matter
(Engel et al., 2004; Kepkay and Johnson, 1989; Mopper et al., 1995; Passow, 2012; Schuster
and Herndl, 1995; Zhou et al., 1998), as well as microbial interactions (Kepkay and Johnson,
1989). This mechanical disturbance may be particularly exacerbated when bubbling is used in
small-volume incubations at the laboratory/microcosm experimental scale (<20 litres). An
alternative method of introducing $CO_2$ gas is by using gas-permeable tubing (Law et al., 2012;
Hoffmann et al., 2013), which eliminates physical artefacts associated with bubbling whilst
achieving realistic future carbonate chemistry. Previous research has been conducted
comparing the effect of acid addition and $CO_2$ gas bubbling on phytoplankton growth, with no
significant effect detected (Chen and Durbin, 1994; Hoppe et al., 2011; Shi et al., 2009).
However, to date no comparison of the bacterial response to seawater acidified with acid and
$CO_2$ gas aeration has been carried out. In addition, there are no published comparisons of $CO_2$
gas introduced through gas-permeable silicon tubing with bubbling to assess their suitability
for OA research. Consequently the aims of the following study were two-fold; to determine the
effect of pH on the sensitivity of fluorogenic substrates used bacterial enzyme analysis, and
also to compare the response of bacterial processes to different approaches of acidification in
small-volume incubations.

## 111   2 Material and methods

### 112   2.1 pH determination

Sample pH was determined using a CX-505 laboratory multifunction meter (Elmetron)
equipped with a platinum temperature integrated pH electrode (IJ44C-HT enhanced series;
accuracy 0.002 pH units), calibrated using Tris buffers (Cornwall and Hurd, 2015) and
regularly cleaned using potassium chloride reference electrolyte gel (RE45-Ionode). Electrode
pH measurements were validated using a pH spectrophotometer with colorimetric
determination using a thymol blue dye solution (Law et al., 2012; McGraw et al., 2010).
Following recommendations in the European Project on Ocean Acidification (Riebesell et al.,
2010), pH values in this research reflect the total hydrogen ion scale ($pH_T$).




## 2.2 Extracellular enzyme activity

The activity of two proteases was examined, with arginine aminopeptidase activity (AAP) quantified using L-arginine-7-amido-4-methylcoumarin hydrochloride (Arg-MCA), and leucine aminopeptidase activity (LAP) quantified using L-leucine-7-amido-4-methylcoumarin hydrochloride (Leu-MCA). Two glucosidases were also examined; α-glucosidase activity (AG) was quantified using 4-Methylumberlliferyl *a*-D-glucopyranoside (α-MUF), and *β*-glucosidase activity (BG) was quantified using 4-Methylumberlliferyl *β*-D-glucopyranoside (*β*-MUF, all from P212121 LLC, USA). Artificial fluorogenic substrate was added to each seawater sample to give a final substrate assay concentration of 39 μM, which was determined from independent tests to be the optimum concentration for calculating the maximum velocity of enzyme hydrolysis in seawater samples (data not shown). A four point calibration curve (0, 4, 40, 200 nM final concentration) was created using 4-methylumbeliferone (MUF) for glucosidase activity, with a separate calibration curve (0, 40, 400, 4000 nM final concentration) created using 7-amino-4-methylcoumarin (MCA) for protease activity (Sigma-Aldrich). UltraPure distilled water (Invitrogen™, Life Technologies) was used as a sample blank. Each sample was assayed in triplicate using a single 96-microwell flat bottom black assay plate (Nunc A/S), with a separate enzyme assay performed for glucosidase and protease activity. Each assay plate was read at 5 min intervals for a minimum of 3 h using a Modulus microplate reader (Turner Biosystems) at 365 nm excitation and 460 nm emission wavelength as in Burrell et al., (2015). Incubation assay temperature matched the seawater temperature at the sampling site. The potential for outgassing and associated increase in sample pH during the 3 h enzyme assay was not tested. The maximum potential enzyme rate ($V_{max}$, nmol $l^{-1}$ $h^{-1}$) was approximated from the saturating substrate concentration of 39 μM. Triplicate $V_{max}$ approximations were averaged per sample. Cell-specific rates were calculated by dividing the activity per litre by bacterial cell numbers per litre. The assay tests were carried out using surface seawater collected from the south coast of Wellington, New Zealand (41°20'53.0"S, 174°45'54.0"E).

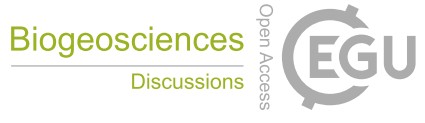



## 2.3 Enzyme assays

### 2.3.1 The effect of pH on fluorophore fluorescence

The effect of pH on fluorophore fluorescence was investigated at both typical (Hoppe, 1983) and elevated fluorophore concentrations using two different buffer solutions, the organic solvent 2-methoxyethanol (Sigma-Aldrich) and 0.1 M Tris/HCl. The pH of MUF and MCA fluorophore working standard (200 μM) diluted in 1 % 2-methoxyethanol (Sigma-Aldrich) was first recorded (pH 6.22 and 6.58 at 18.6 °C respectively). Each fluorophore was then diluted to 4000, 20000 and 40000 nM (referred to as high concentrations) at four pH values (8.2, 8.1, 7.9 and 7.8) in triplicate by addition of 0.1 N aqueous NaOH. The MUF and MCA fluorophore working standards made up in in 0.1 M Tris/HCl were prepared at pH 8.1 and 7.8 only, and also carried out at lower concentrations (MUF: 4, 40, 200 nM; MCA: 40, 400, 4000 nM).

### 2.3.2 The effect of artificial fluorogenic substrate on seawater pH

Individual seawater samples were adjusted to pH 7.95 and 7.70 using 0.1 M HCl. All four artificial fluorogenic substrates previously described were made up to working standards using 1 % 2-methoxyethanol (Sigma-Aldrich). A time-zero reference pH was recorded for each seawater sample and, following the addition of each substrate at 39 μM final concentration, sample pH was recorded immediately and after 30 min. Each artificial fluorogenic substrate was run in triplicate at both pH values, and compared to controls without substrate addition at both pH levels.

### 2.3.3 Buffering artificial substrates

Duplicate trials were undertaken to determine if 0.1 M Tris/HCl could successfully buffer MCA substrate at the working concentration (39 μM) when added to seawater of similar pH. Tris buffer contains an amine group which can affect peptidase activity (Baker and Prescort, 1983; Desmarais et al., 2002; Saishin et al., 2010), and so tests were carried out to compare the impact of different buffers. LAP activity was compared in seawater using LAP substrate (39 μM final concentration) buffered with 0.1 M Tris/HCl or 3-(N-morpholino)propanesulfonic acid (MOPS) with pH adjusted to 8.1. Enzyme activity was also determined in seawater (pH 8.18). A non-buffered LAP substrate addition was not included due to the acidic nature of the aminopeptidase substrate (non-buffered LAP substrate was pH 5.87). MOPS has been used as a buffer in studies of the effects of pH on enzymes (Piontek et al. 2010), and so was an



appropriate comparison. Borate buffers were not trialled because they have a bactericidal effect
on microbial activity (Houlsby et al., 1986). In two separate test experiments using coastal
seawater Tris/HCl buffer did not inhibit LAP activity relative to MOPS but instead showed a
minor stimulatory effect with 16-18% higher LAP activity (data not shown). Tris/HCl was
selected for subsequent use as its optimal buffer range is pH 7.8-9.0, making it ideal for OA
incubations, and it has a pKa of 8.06, so is appropriate for artificial fluorescent substrates
(Hoppe, 1993).
Following the above tests, the following methodology was used for the seawater acidification
tests. Tris buffered Leu-MCA and Arg-MCA substrate working standards were made by
diluting 500 µl of MCA substrate stock (16 mM) with 4.5 ml of 0.1 M Tris/HCl buffer.
Duplicate Tris/MCA substrate solutions were adjusted to pH 8.1 and 7.8 by adding 10 % HCl
and the pH of duplicate 10 ml aliquots of coastal seawater was also adjusted to pH 8.1 and 7.8.
For each pH treatment, 250 µl of Tris/MCA substrate solution was added to 10 ml of seawater
fixed at the corresponding pH. pH was recorded at room temperature using a pH electrode as
described above.

## 2.4 Seawater acidification approach

The influence of acidification technique on biotic parameters was investigated in two separate
experiments conducted under controlled temperature conditions in late summer (May 2013 -
trial 1) and in early spring (October 2013 - trial 2). Coastal seawater was first filtered through
a 15 µm filter and then a 1 µm inline cartridge filter. Three different methods were used to
acidify seawater to that predicted by the end of the century (pH 7.80) (IPCC, 2013): (A) acid
addition using 0.1 M HCl; (B) bubbling $CO_2$-Air gas mixture through an acid-washed aquarium
airstone, and (P) $CO_2$-Air gas mixture introduced through gas-permeable silicon tubing (Tygon
Tubing R-3603; ID 1.6 mm; OD 3.2 mm; Law et al, 2012). Treatment P was acidified to a pH
of 7.8 by the sequential application of 100 % synthetically produced $CO_2$ gas for 25 min,
followed by 10 % $CO_2$ gas (in 20.8 % $O_2$ in $N_2$, BOC Gas Ltd) for 60 min at a flow rate of <
26 ml min$^{-1}$. The initial use of pure and 10 % $CO_2$ gas made it possible to reach the target pH
within 3 h. Treatment B was acidified by bubbling seawater with 742 µatm $CO_2$ gas (in 20.95
% $O_2$ in $N_2$, BOC Gas Ltd) for 143 min at < 25 ml min$^{-1}$ to achieve the target pH 7.80. The
volume of 0.1 M HCl required to acidify treatment A to pH 7.8 (2.0 ml - trial 1, 3.1 ml - trial



2) was calculated based on the sample volume, DIC and alkalinity (pers. comm. Dr K. Currie,
NIWA/University of Otago) using an algorithm from Dickson et al. (2007). To ensure a
consistent rate of pH change across treatments, treatment B and A were adjusted to match that
of the slower treatment P (150 min), with the pH of each sample verified using a pH electrode.
Each treatment and an ambient seawater Control were then incubated in triplicate in acid-
washed milli-Q water-rinsed 4.3 Litre low-density polyethylene (LDPE) cubitainers
(ThermoFisher Scientific), without a headspace. No further pH adjustment took place during
the 96 h incubation.
Each cubitainer was housed in one of two identical perspex incubation chambers (1730 mm
long, 450 mm high by 325 mm deep), set at *in situ* ambient seawater temperature (15.1 ℃ -
trial 1, 15.5 ℃ - trial 2). Artificial light (700 - 900 $\mu E\,m^{-2}\,s^{-1}$) was maintained in each cubitainer
through external fluorescent light banks (Philips TLD 36 W/840); neutral density
polycarbonate screening ensured light intensities were uniform between incubation chambers,
while adjustable timers ensured an automated diurnal 12 h light/dark cycle. Mixing of water in
each cubitainer was achieved using an inflating diaphragm positioned underneath each
cubitainer, with the inflation and collapse of the diaphragm under the weight of the sample
resulting in continual water mixing. Cubitainers were also manually removed and inverted
three times prior to each sampling. Time-zero sampling occurred after initial pH adjustment.
Assay fluorophore and substrate standard solutions were adjusted to treatment pH.

### 2.4.1 Bacteria and picoplankton cell numbers

Triplicate samples were collected in 2 ml Cryovials (Raylab Ltd) and frozen in liquid nitrogen
(Hall et al., 2004) for up to 12 weeks prior to analysis. Bacterial cell numbers were determined
by flow cytometry (FACSCalibur, Becton-Dickinson) following staining with SybrGreenII
(Invitrogen) (Lebaron et al., 1998), and count events were normalised to volume using
TruCount bead solution (BD Biosciences) (Button and Robertson, 1993). Total eukaryotic
picoplankton numbers (< 2 $\mu$m) were determined by fluorescence of chlorophyll (wavelength
670 nm), phycoerythrin (585 nm), and phycourobilin (530 nm) as well as forward light-scatter
providing an estimate of cell size. Final count values were recorded as cells $ml^{-1}$.

### 2.4.2 Bacterial secondary production

Potential bacterial secondary production (BSP) was measured using $^3$H-leucine ($^3$H-Leu) of
high specific activity (> 80 Ci $mmol^{-1}$, SciMed Ltd) in triplicate 1.7 ml samples. Following the





TCA precipitation and centrifugation methodology (Kirchman, 2001; Smith and Azam, 1992),
$^{3}$H-Leu incorporation was determined using a liquid scintillation counter (Tri-Carb 2910 TR)
and converted to secondary production using a protein conversion factor (1.5 kg C mol$^{-1}$
leucine) (Simon and Azam, 1989). Cell-specific rates were calculated by dividing the BSP rate
by total bacterial cell numbers.

### 2.4.3 Dissolved Inorganic Carbon and Total Alkalinity

Pre-combusted 12 ml sample DIC vials (Labco Ltd) were triple rinsed with sample seawater
and filled, ensuring no air bubbles. One drop of saturated HgCl$_2$ was added to each DIC sample,
with storage at room temperature. DIC was determined using evolved CO$_2$ gas after sample
acidification on a Marianda AIRICA system, the accuracy of this method was estimated to be
$\pm 5$ µmol kg$^{-1}$, as determined by analysis of Certified Reference Material. Alkalinity samples
were collected by filling a 1 liter screw top bottle, and following the same sample preparation
and storage procedures as DIC above. Samples were later analysed by potentiometric titration
in a closed cell (Dickson et al., 2007) with an accuracy of $\pm 2$ µmol kg$^{-1}$, also determined by
analysis of Certified Reference Material.

### 2.5 Statistical analysis

Statistica v.10 (StatSoft Inc., USA) was used for basic graphics and descriptive statistics. Data
was tested for normality and equality of variance prior to statistical analysis. Data was log(x+1)
transformed due to the small sample size at each sampling point. Standard hypothesis
formulations were used for each Analysis of Variance (ANOVA), the null hypothesis (H$_o$) was
$\mu = 0$. The significance level of each test was $p \leq 0.05$. If H$_o$ was rejected, a Tukey's HSD post-
hoc analysis test was run to identify individual variable responses.

## 3 Results and discussion

### 3.1 Enzyme assay methodology

MUF and MCA fluorescence was lower at pH 7.8 relative to pH 8.1, as previously reported in
soils (Niemi and Vepsäläinen, 2005). The fluorescence of the unbuffered MUF 2-
methoxyethanol at 40000 nM was 20 % higher at pH 8.1 than at pH 7.8 (t-test, $p < 0.05$), while
MUF Tris buffered fluorescence at 200 nM was 15 % higher at pH 8.1 (t-test, $p < 0.05$; Table
1). MCA 2-methoxyethanol fluorescence at 40000 nM was 4 % higher at pH 8.1 than



fluorescence at pH 7.8 (t-test, $p < 0.05$), while MCA Tris buffered fluorescence at 200 nM was
9 % higher at pH 8.1 than at pH 7.8 (t-test, $p < 0.05$; Table 1). These results confirm that pH
has a significant effect on MUF and MCA fluorescence at both high and typical working
concentrations, and so fluorophore calibrations should be carried out at the same pH as the
sample.
Although there is awareness of the effect of pH on fluorophore fluorescence (Piontek et al.,
2013; Endres et al., 2014), few studies consider the effect of fluorescent substrate addition on
seawater pH. Due to the basicity of the MCA amino group, fluorescence intensity is less
affected by pH and it has been suggested that buffering is not required (Piontek et al., 2013;
Endres et al., 2014), whereas buffering of MUF has been reported (Piontek et al., 2010; 2013,
Endres et al., 2013). Immediately following the addition of non-buffered Leu-MCA or Arg-
MCA substrate to seawater at pH 7.95 or 7.70, pH decreased by at least 0.05 units for each
substrate, and remained significantly lower 30 mins after addition when compared to time-zero
pH (one-way ANOVA, $p < 0.05$). As both MCA substrates are hydrochloride salts, addition
resulted in a significant pH change, as previously reported by Hoppe (1993). In tests of Tris
buffered MCA substrate solutions adjusted to seawater pH 7.8 and 8.1, pH change ranged from
0.003 to 0.03 units (±0.001 SE). As the addition of buffer solution reduced the pH change, both
MCA substrates and fluorophores were subsequently produced using 0.1 M Tris/HCl, with pH
adjusted to the respective experimental treatments and Control. In contrast to MCA, no
statistically significant change in pH was recorded immediately following, or 30 mins after,
addition of either α-MUF or β-MUF substrate to seawater at pH 7.95 or 7.70, indicating that
these are neutral compounds. However, to eliminate possible bias, MUF substrates were also
buffered using Tris/HCl.

**3.2 Seawater acidification**
Having established that the analytical procedures for determining extracellular enzyme activity
are affected by, and alter pH, the influence of acidification technique was then considered in
two separate trials in different seasons. Overall, the experiments showed that different
acidification techniques had significant effects on BG and LAP activity in both trials (Fig. 1),
while the response of AG and AAP activity was variable with no consistent treatment response
relative to the Control (data not shown). Overall, BG and AG activity declined from time-zero



to 96 hrs in the Control and treatments in trial 1, but were both significantly higher in the
treatments relative to the Control from time-zero to 72 h, with BG activity approximately three-
fold higher than AG activity (data not shown). Cell-specific BG activity was at least an order
of magnitude higher in treatment B, P and A relative to the Control at time-zero (one-way
ANOVA, $p < 0.05$) (Fig. 2), which is consistent with a direct effect of acidification (Piontek et
al., 2013). Cell-specific BG activity was highest in treatment B from 24 h to 72 h by at least 14
% relative to treatment A and P (Fig. 1). In contrast to trial 1, cell-specific BG activity increased
significantly throughout trial 2 (repeated measures ANOVA, $p < 0.05$). The opposing temporal
trends between trials may signify seasonal differences in the response of glucosidase to OA,
potentially reflecting differences in microbial community composition (Endo et al., 2013) or
substrate availability (Morris and Foster, 1971). There was no significant difference in BG
activity between treatments at time-zero in trial 2 (one-way ANOVA, $p > 0.05$) (Fig. 2), and
BG activity was again highest in treatment B from 48 h, with activity at least 18 % higher
relative to treatment P and A (Fig. 1). Bulk water LAP and AAP activity varied between
treatments for trials 1 and 2. For example, both LAP and AAP activity were highest in treatment
P throughout trial 1, whereas LAP activity was highest in treatment B from 72 h to 96 h in trial
2 (data not shown). Although cell-specific LAP activity showed evidence of a response to
acidification, this was not significant in either trial (Fig. 1).
Although treatment B was only bubbled with gas mixtures for the pre-incubation period (143
mins), this had a greater effect on BG activity than in the other treatments, indicating potential
artefacts associated with bubbling. Bubbling may have ruptured picoplankton cells or increased
their susceptibility to viral lysis, leading to an increase in the release of labile organic
carbohydrates. This is potentially supported by the decline in total eukaryotic picoplankton cell
numbers in treatment B (trial 1 – 2.8 x $10^3$ to 2.6 x $10^3$ cells ml$^{-1}$, trial 2 – 1.7 x $10^3$ to 1.3 x $10^3$
cells ml$^{-1}$) in both trials (repeated measures ANOVA, $p < 0.01$). An increase in enzyme activity
would theoretically increase the availability of low molecular weight organic substrate for
bacterial assimilation, and may explain the significant increase in bacterial cell numbers in
treatment B relative to the Control at 96 h in both trials (one-way ANOVA, $p < 0.05$) (Fig. 2).
An increase in bacterial abundance in response to bubbling has been previously reported by
(Kepkay and Johnson, 1989) who suggested that surface DOC coagulation facilitated by
bubbling resulted in increased respiration and bacterial numbers. It is possible that bubbling
increased the abiotic coagulation of organic matter (Riley, 1963) and formation of high



molecular weight substrate such as transparent exopolymer particles (Mopper et al., 1995;
Passow, 2012; Schuster and Herndl, 1995; Zhou et al., 1998), which could explain the elevated
cell-specific BG activity (Fig. 1).
All acidification treatments had a significant negative effect on cell-specific BSP from 24 h to
48 h in trial 1 (one-way ANOVA, $p < 0.05$) (Fig. 3). During trial 2, cell-specific BSP was
significantly lower in treatments B and P when compared to the Control from 72 h to 96 h (one-
way ANOVA, $p < 0.05$), while BSP was twice as high in treatment A during this period (Fig.
3). Although a clear treatment response was not observed in either trial, the low cell-specific
BSP in treatment B relative to the Control and treatment A at 96 h in trial 2 was surprising as
enzyme activity and bacterial cell numbers were elevated. Existing literature also reports
variable BSP responses to acidified conditions. Arnosti et al., (2011) and Teira et al., (2012)
detected no significant BSP response, while Grossart et al., (2006) detected an increase, and
Maas et al., (2013) and Siu et al., (2014) recorded a decrease in BSP rates with increasing $CO_2$.
As the same response was not observed in trial 1, it is possible that additional indirect factors
such as bacterial community composition or substrate type may have influenced BSP under
OA conditions (Piontek et al., 2013).

## 4 Conclusions

Artificial fluorogenic substrates have been used to investigate bacterial extracellular enzyme
activities in aquatic environments for decades (Hoppe, 1983; Somville and Billen, 1983).
Although the technique has several limitations, including that the artificial fluorogenic
substrate may not represent the naturally occurring substrate (Chróst, 1989), and so the
observed activity only represents potential hydrolysis (Arnosti, 1996; Unanue et al., 1999), the
technique is rapid and easily applied in the field and most importantly, allows for a standardised
method for comparison of results in different OA studies. This study confirmed that artificial
fluorogenic substrates used to determine extracellular enzyme activity are affected by, or alter,
pH, and so buffering is required particularly when used in OA research. Seawater acidification
stimulated $\beta$-glucosidase activity, but different methodological approaches can influence the
magnitude of this response. Simple acid addition does not produce realistic seawater carbonate
chemistry predicted in a future ocean (Riebesell et al., 2010), and bubbling with $CO_2$ gas has a
significant effect on $\beta$-glucosidase activity and bacterial cell numbers, indicating that there are



artefacts associated with bubbling. It should be noted that these effects were observed in small-
volume laboratory-scale experiments, and may have less impact in larger-scale experiments.
Nevertheless, the results indicate that the most robust technique to investigate the response of
bacterial processes to future OA conditions is $CO_2$-Air gas mixtures introduced using gas
permeable-silicon tubing. This approach should be considered for broader use in standardised
protocols for ocean acidification (Riebesell et al., 2010; Cornwall and Hurd, 2015) to achieve
robust meta-analyses and international inter-comparisons.

**Acknowledgements**
This research was supported by a Marsden Fund Award from New Zealand Government
funding, administered by the Royal Society of New Zealand to E. W. Maas and C. S. Law. We
acknowledge assistance from Kim Currie, Debbie Hulston, Marieke van Kooten, Cara Mackle
and Karen Thompson. We also thank John van der Sman for seawater supplied by the Victoria
University Coastal Ecology Laboratory, Wellington.



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



**Figure Legends**
Table 1. Mean fluorophore fluorescence at pH 8.1 ad 7.8 (RFU, n=3, ±SE).

|  | *Concentration (nM)* | *Fluorophore* | *pH 8.1* | *pH 7.8* |
|---|---|---|---|---|
| *0.1M Tris* | *200* | *MUF* | *1621.44 (±3.43)* | *1373.33 (±2.49)* |
|  |  | *MCA* | *14948.90 (±2.52)* | *13626.54 (±2.52)* |


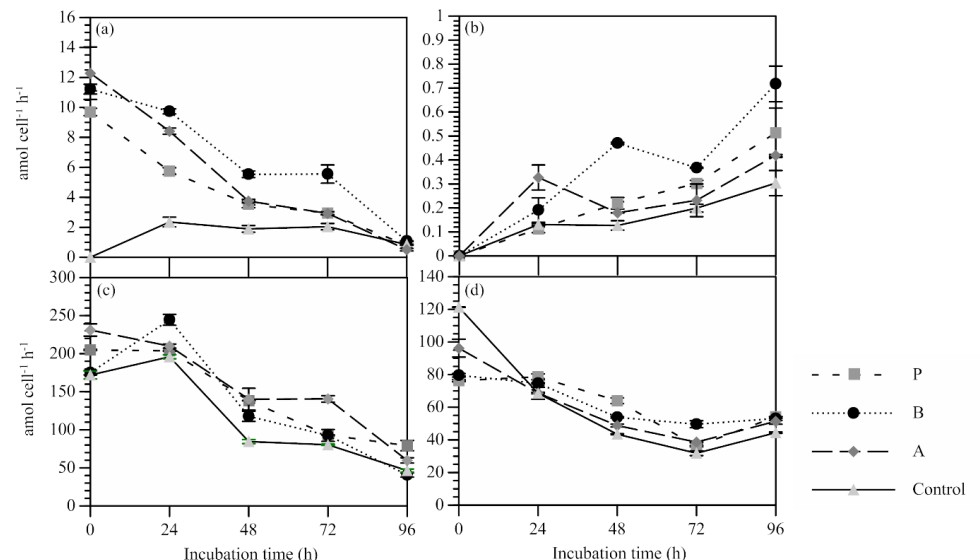


Figure 1. Cell-specific extracellular enzyme activity (mean ± SE, n=3) in response to seawater
acidified with 0.1 M HCl (A), bubbled with $CO_2$-Air gas mixture (B) and $CO_2$-Air gas mixture
introduced through gas-permeable silicon tubing (P). (a) BG activity in trial 1, (b) BG activity
in trial 2, (c) LAP activity in trial 1, (d) LAP activity in trial 2.







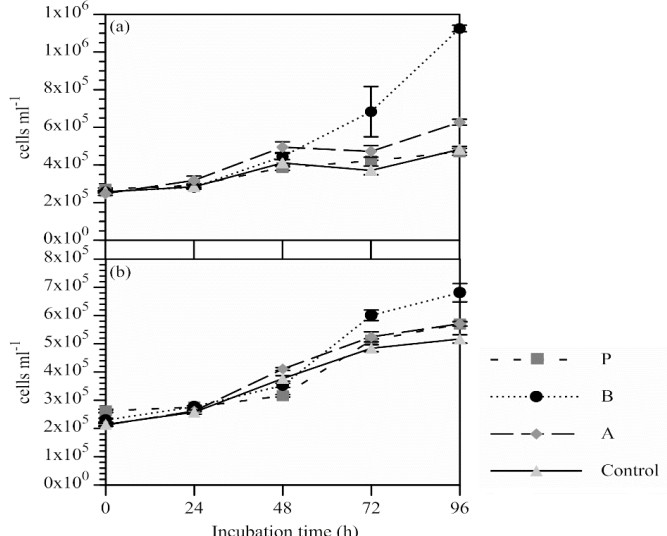


Figure 2. Bacterial cell numbers (mean ± SE, n=3) in response to seawater acidified with 0.1 M HCl (A), bubbled with $CO_2$-Air gas mixture (B) and $CO_2$-Air gas mixture introduced through gas-permeable silicon tubing (P). (a) trial 1, (b) trial 2.

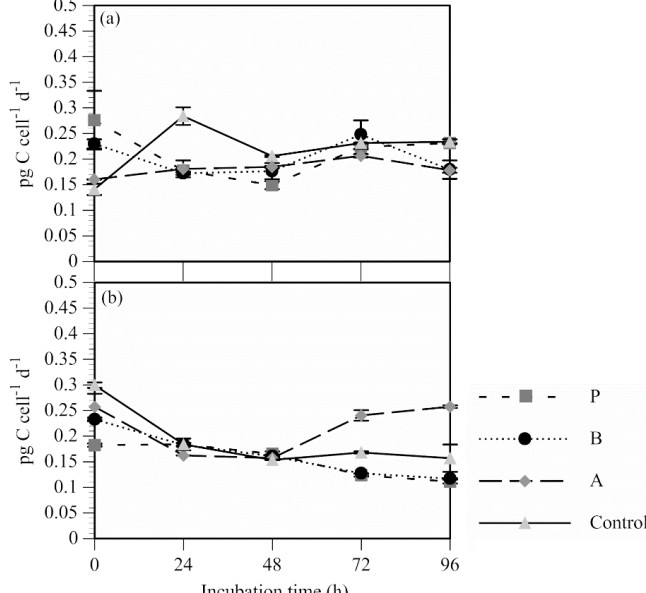


Figure 3. Cell-specific bacterial secondary production (mean ± SE, n=3) in response to seawater acidified with 0.1 M HCl (A), bubbled with $CO_2$-Air gas mixture (B) and $CO_2$-Air gas mixture introduced through gas-permeable silicon tubing (P). (a) trial 1, (b) trial 2.