# Peer review of "Assessing approaches to determine the effect of ocean"

_Biogeosciences, 2016_

## Referee Comment (RC1) · Anonymous Referee #1 · 19 Apr 2016

The manuscript addresses a well-defined question that is relevant for the interpretation and comparison of results of various experiments conducted with the objective of predicting the impact of ocean acidification on bacterial processes of organic matter degradation and recycling. The introduction provides a good context to the problem. Methodologies are well described and the results are clearly presented. The conclusions are solid, well based on the critical interpretation of results and likely to be taken into account in the planning of new experiments. Specific comments: The effect of pH on MUF florescence is well known and 4-methylumbelliferone was even proposed as an acid-base fluorescent indicator (Chen, 1968). Because of this, sometimes a glycine-ammonium buffer pH 10.5 is added just before measuring fluorescence. This enhances MUF fluorescence without affecting the biological processes. Would this approach attenuate the effect of pH on the estimated results? Fluorescence was de-

termined at the same wavelengths for MUF and MCA but the fluorescence spectra of the two molecules are different. For MCA, 380 nm excitation and 440 nm emission are often used. Can this slight difference have any effect on the results and conclusions?

---

## Referee Comment (RC2) · Anonymous Referee #2 · 18 May 2016

The authors studied the response of bacterial growth, production and hydrolytic activity to acidification comparing three common acidification methods used in ocean acidification research. They conclude that magnitude and direction of the response may depend on the type of manipulation (acid addition, bubbling, gas-permeable tubing). Unfortunately, they did not evaluate two frequently used methods in ocean acidification research: the addition of high $CO_2$/supersaturated seawater and the addition of bicarbonate. Nevertheless, the effect of the acidification methods on bacterial processes has not been compared so far. The presented data are mostly novel and interesting and the subject area is clearly appropriate for publication in Biogeosciences. The experiments were correctly planned, described and thoroughly carried out. For these reasons I think that the manuscript deserves publication in Biogeosciences. There are, however, a couple of points of relatively major nature – especially in discussion and

conclusion section - that the authors should take into account in a revised version of the manuscript:

The title "Assessing approaches to determine the effect of ocean acidification on bacterial processes" implies a broader perspective than provided by the authors. As a reader, I would expect an assessment of OA methods on various bacterial processes such as growth, secondary production and degradation of organic matter. In fact, those processes were studied and data are provided (although only briefly), but unfortunately not discussed in this context. In my opinion, the authors' focus is to narrow on extracellular enzyme activities and substrate fluorescence. I suggest broadening the discussion of the results including OA effects on bacterial growth and production and changing the conclusion section as well as the abstract accordingly. The authors could also speculate in their discussion whether acidification may have influenced enzyme expression or lifetime therefore indirectly affecting enzyme rates.

The authors studied the effect of pH on substrate fluorescence (MUF and MCA) as well as the effect of substrate addition on seawater pH. The addition of MCA affected seawater pH, while the addition of MUF didn't. This is a very interesting result and to my knowledge has not been shown before. The authors should stress their point that buffering is necessary when determining enzyme rates in general, or at least when using MCA as a marker. In contrast, the effect of pH on MUF fluorescence is well known (e.g. Mead et al. 1955), explicitly written in the Sigma product information and usually considered in enzyme rate measurements. Furthermore, I am not convinced by the authors' proposed effect of pH on MCA fluorescence (see specific comments below).

Mead, J. A. R., et al., The biosynthesis of the glucuronides of umbelliferone and 4-methylumbelliferone and their use in fluorimetric determination of beta-glucuronidase. Biochem. J., 61, 569-574 (1955).

Specific comments:

p. 1 l. 17f: Change to "This study investigated the potential artefacts in determining the response of bacterial growth and activity to ocean acidification, and the relative effects of three different acidification techniques."

p. 1 l.26ff: From the presented results I would conclude that "bubbling may stimulate carbohydrate degradation and bacterial growth".

p. 2 l.32ff Add some more information on extracellular enzyme characteristics: Enzymes are considered as the rate limiting step in hydrolysis of HMW-substrate by bacteria. Both enzyme groups consist of several isoenzymes that catalyze the same reaction but may vary significantly in e.g. pH or temperature optimum and sensitivity (e.g. broad range or narrow optimum range). Define" extracellular enzyme". Do you include cell-attached and particle-attached enzymes or only free enzymes?

p. 2 l.56: What are "indirect influences on longer timescales"? Please specify.

p. 3 l. 66ff: The pH sensitivity of MUF is well known (e.g. Mead et al. 1955 and SIGMA product information). Please clarify this in the text.

p. 4 l. 107: see above

p. 5 l. 130ff: It would be very interesting to see the kinetic curves of the independent tests that the authors mention. Please provide a short table or graph. Enzyme kinetics and maximum velocities may vary from one seawater sample to the other (depending on isoenzymes present in the sample). Did you test enzyme kinetics both, in summer and spring?

p. 5 l. 132f: At which pH did you calibrate MUF?

p. 7 l. 182: Please include data (e.g. as supplementary graph)

p. 8 l. 216: Did you determine pH at the end of the incubations?

p. 8 l. 220: Was there a reason to incubate under artificial light instead of dark incubations?

p. 9 l. 270ff: Can you please give the standard deviation of your calibration? An increase by 4% only seems to me very low and within the experimental detection limit. Previous studies did not detect a significant effect of pH on MCA fluorescence and I would not consider 4% to be significant. It would be useful, if you could provide a graph with the calibration curves of both fluorescent markers at pH 8.1 and pH 7.8!

p. 10 l. 299 I agree that different acidification methods had significant effects on BG activities, but I cannot see a significant effect on LAP activity from the presented data (Figure 1). In p. 11 l. 318 the authors state that, although cell-specific LAP activity showed evidence of a response to acidification, this was not significant in either trial. Please explain/clarify and give statistical evidence. It would be also interesting to see the data for AG and AAP activity.

p. 12 l. 337ff: What about total secondary production rates? How do you explain difference towards the end of trial 2? Can you relate it to changes in BG activity?

p. 13 l. 367: The authors state that the introduction of CO2-air gas mixtures using gas-permeable tubing would be the "most robust technique to investigate the response of bacterial processes to future OA conditions". This is ignoring the fact that there are more techniques commonly used which were not tested in this study and may be even "more robust". Furthermore, I would conclude from this study that different techniques may result in different results. They may under- or overestimate certain parameters at the same time but not all parameters equally.

---

## Author Response (AR1)

**Author response to comments from anonymous referee #1 - manuscript bg-2016-63**

We thank the reviewer for their constructive comments. Below please find our point by point responses to the referees' specific comments.

1) *The reviewer asks whether using a glycine-ammonium buffer (pH 10.5) would be beneficial prior to measuring fluorescence.*

   As the reviewer notes, the pH effect for the MUF substrate was previously known. The objective of our research was to investigate if there was a pH dependency in the MCA assay, and whether our assay required buffering. We have shown that for this substrate it is important to control pH. We wished to maintain a constant and defined pH throughout the incubation as well as the fluorescence measurement, as the specific activity of enzymes' can vary with pH. Thus we favoured buffering at a pH relevant to enzyme activity, rather than allow pH to "roam" during the incubation, with potential variations in product formation between experiments.

2) The reviewer notes that f*luorescence was determined at the same wavelengths for MUF and MCA, but the fluorescence spectra of the two molecules are different. The review asked whether this slight difference could have any effect on the results and conclusions?*

   For robustness we used filter blocks in our sea-going plate reader. 365 nm excitation and 460 nm emission wavelengths was the only block available that covered the correct wavelengths. Although we may be off the ideal excitation and emission wavelengths, which raises our limit of detection, this should not alter the results or conclusions, unless there was something else in the system giving an emission at that wavelength and whose emission was pH responsive. We ran trials with natural seawater to ensure there was no inherent interference with fluorescence. Variation in the wavelengths used for MUF & MCA also exists within current literature. For instance, Chrost (1992) use the same MUF excitation and emission as we report, but used 380 nm excitation and 440 nm emission for MCA (as stated by the reviewer), while Hoppe et al (1988) used 365 excitation and 445nm. Christian & Karl (1995) used 360 nm excitation and 447nm emission for MUF, while both Mass et al. (2013) and Piontek et al. (2009, 2010, 2013) uses 355nm excitation and 460 nm emission for both MUF & MCA. In our study, it was important to use wavelengths used by others for consistency and comparison of responses.

**Author response to comments from anonymous referee #2 - manuscript bg-2016-63**

We thank the reviewer for their constructive comments. Below please find our point by point responses to the referees' specific comments.

*In my opinion, the authors' focus is to narrow on extracellular enzyme activities and substrate fluorescence. I suggest broadening the discussion of the results including OA effects on bacterial growth and production and changing the conclusion section as well as the abstract accordingly.*

We have broadened the discussion to include additional bacterial responses to OA. The conclusion and abstract have been amended accordingly.

*The authors should stress their point that buffering is necessary when determining enzyme rates in general, or at least when using MCA as a marker.*

We have further emphasized this observation.

*In contrast, the effect of pH on MUF fluorescence is well known (e.g. Mead et al. 1955), explicitly written in the Sigma product information and usually considered in enzyme rate measurements. Furthermore, I am not convinced by the authors' proposed effect of pH on MCA fluorescence.*

We acknowledge previous studies on the effect of pH on MUF fluorescence, and have revised our interpretation on the effect of pH on MCA fluorescence (see below).

*Specific comments:*

*p. 1 l. 17f: Change to "This study investigated the potential artefacts in determining the response of bacterial growth and activity to ocean acidification, and the relative effects of three different acidification techniques."*

This has been changed in the revised manuscript.

*p. 1 l.26ff: From the presented results I would conclude that "bubbling may stimulate carbohydrate degradation and bacterial growth".*

This has been changed in the revised manuscript.

*p. 2 l.32ff Add some more information on extracellular enzyme characteristics: Enzymes are considered as the rate limiting step in hydrolysis of HMW-substrate by bacteria. Both enzyme groups consist of several isoenzymes that catalyze the same reaction but may vary significantly in e.g. pH or temperature optimum and sensitivity (e.g. broad range or narrow optimum range). Define" extracellular enzyme". Do you include cell-attached and particle-attached enzymes or only free enzymes?*

Additional information has been incorporated as requested. Yes, the presented enzyme activities also include any cell- or particle attached enzyme activity. The definition of 'extracellular' in the context of this work has been clarified in the revised manuscript.

*p. 2 l.56: What are "indirect influences on longer timescales"? Please specify.*

We are referring to potential indirect effects on enzymatic rate, due to factors including altering enzyme abundance, changes in substrate arising from plankton community composition change, grazing or viral lysis, abiotic influences on rate (e.g. through pH effects or carbamylation), and type of enzyme synthesized, as opposed to direct effects of change in pH on the enzyme activity. This has been clarified in the revised manuscript.

*p. 3 l. 66ff: The pH sensitivity of MUF is well known (e.g. Mead et al. 1955 and SIGMA product information). Please clarify this in the text. p. 4 l. 107: see above*

This has been clarified in the revised manuscript.

*p. 5 l. 130ff: It would be very interesting to see the kinetic curves of the independent tests that the authors mention. Please provide a short table or graph. Enzyme kinetics and maximum velocities may vary from one seawater sample to the other (depending on isoenzymes present in the sample). Did you test enzyme kinetics both, in summer and spring?*

The tests to calculate the optimum substrate concentrations were conducted by another researcher using seawater collected from a similar location. We agree that enzyme kinetics and maximum velocities may vary spatially and temporally; however, as the purpose of this study was to assess enzyme response to acidification, and to different methods of acidification, full enzyme kinetics were not required. We believe that the variance between samples collected from different sites acts as a reasonable proxy for the variance that would be found between different seasons at the same site.

*p. 5 l. 132f: At which pH did you calibrate MUF?*

At pH 7.8 and 8.1. This has been clarified in the revised manuscript.

*p. 7 l. 182: Please include data (e.g. as supplementary graph)*

As requested, the following table compares average LAP activity buffered in 0.1 M Tris and MOPS at pH 8.1 in coastal seawater. This has been added as supplementary material (S1) in the revised manuscript.

| Trial | Tris activity (nmol $l^{-1}$ $h^{-1}$) | MOPS activity (nmol $l^{-1}$ $h^{-1}$) |
|---|---|---|
| 1 | 51.54 (±2.32) | 43.42 (±1.43) |
| 2 | 35.92 (±0.81) | 29.34 (±1.08) |

*p. 8 l. 216: Did you determine pH at the end of the incubations?*

pH was determined at 0, 24, 48, 72 and 96 h in both trials. This data has been added to supplementary material (S2 and S3).

*p. 8 l. 220: Was there a reason to incubate under artificial light instead of dark incubations?*

The aim was to simulate natural conditions in coastal waters.

*p. 9 l. 270ff: Can you please give the standard deviation of your calibration? An increase by 4% only seems to me very low and within the experimental detection limit. Previous studies did not detect a significant effect of pH on MCA fluorescence and I would not consider 4% to be significant. It would be useful, if you could provide a graph with the calibration curves of both fluorescent markers at pH 8.1 and pH 7.8!*

We were prompted by the reviewer's concern to refer back to this data and consider other data not previously interpreted for this paper. We find a similar trend in this revised data set, (shown in revised Table 1), but not an acceptable level of significance ($p > 0.05$). We thank the reviewer for raising the issue, and in the light of this more recent analysis have revised this statement to clearly state that there is no robustly significant difference in the fluorescence of Tris buffered MCA or MUF between 7.8 and 8.1. This result is included in the abstract.

*p. 10 l. 299 I agree that different acidification methods had significant effects on BG activities, but I cannot see a significant effect on LAP activity from the presented data (Figure 1). In p. 11 l. 318 the authors state that, although cell-specific LAP activity showed evidence of a response to acidification, this was not significant in either trial. Please explain/clarify and give statistical evidence.*

We agree with the reviewers comment that this sentence is confusing and requires rewording. The reference to "significant effects' in Line 299 refers to the response of different treatments at select sampling points only. For instance, activity in treatment A was significantly higher than the Control at 48 and 72 h during trial 1 (t-test, df=4, p=<0.001) (Fig 1), while activity in treatment B was frequently higher than the Control in both trials however was not significantly greater. So although there was 'evidence' of a response to acidification, these were not statistically significant. This has now been clarified in the revised manuscript.

*It would be also interesting to see the data for AG and AAP activity.*

Trial 1 AG and AAP activity, and trial 2 AAP activity have now been included in the supplementary material (S4, S5, and S6 respectively). AG activity was very low during trial 2, being below detection at several sampling points across all treatments; consequently we mention this, but do not include the data in the supplementary material.

*p. 12 l. 337ff: What about total secondary production rates? How do you explain the difference towards the end of trial 2? Can you relate it to changes in BG activity?*

The response in secondary production in the latter part of trial 2 (Fig. 3) does not reflect the response in BG activity in the different treatments (Fig. 1). The increase in BG activity is highest in treatment B relative to the other treatments, while the increase in secondary production occurs in treatment A. The response in secondary production observed from 72 h during trial 2 may be explained by other indirect factors, as discussed on l. 355.

*p. 13 l. 367: The authors state that the introduction of CO2-air gas mixtures using gas-permeable tubing would be the "most robust technique to investigate the response of bacterial processes to future OA conditions". This is ignoring the fact that there are more techniques commonly used which were not tested in this study and may be even "more robust". Furthermore, I would conclude from this study that different techniques may result in different results. They may under- or overestimate certain parameters at the same time but not all parameters equally.*

We agree and have now highlighted the fact that this study only investigates some of the commonly used methods to artificially acidify seawater. We have altered the abstract and conclusion accordingly.

**Assessing approaches to determine the effect of ocean acidification on bacterial processes**

**Tim J. Burrell**[1,2, †]**, Elizabeth W. Maas**[1,*]**, Paul Teesdale-Spittle**[2] **and Cliff S. Law**[1,3]

[1] {National Institute of Water and Atmospheric Research, Greta Point, Wellington, New Zealand}

[2] {Victoria University of Wellington, School of Biological Sciences, Wellington, New Zealand}

[†] {now at: C-MORE, University of Hawaii at Mānoa, Honolulu 96822, Hawaii}

[*] {now at: Ministry for Primary Industry, PO Box 12034, Ahuriri, Napier, New Zealand}

[3] {Department of Chemistry, University of Otago, Dunedin, New Zealand}

Correspondence to: T. Burrell (timbo.burrell@gmail.com)

**Abstract**

Bacterial extracellular enzymes play a significant role in the degradation of labile organic matter and nutrient availability in the open ocean. Although bacterial production and extracellular enzymes may be affected by ocean acidification, few studies to date have considered the methodology used to measure enzyme activity and bacterial processes. This study investigated the potential artefacts in determining the response of bacterial growth and activity bacterial and extracellular glucosidase and aminopeptidase activity to ocean acidification, and the relative effects of three different acidification techniques. Tests confirmed that the observed effect of pH on fluorescence of the artificial fluorophores was affected by pH, and that the influence of addition of the MCA fluorescent substrate on alters seawater sample pH, were both negatedovercome 
[revised manuscript text omitted]